# A map of single-phase high-entropy alloys

Wei Chen [1], Antoine Hilhorst [2], Georgios Bokas[1], Stéphane Gorsse [3], Pascal J. Jacques [2] & Geoffroy Hautier [1,4] ✉

High-entropy alloys have exhibited unusual materials properties. The stability of equimolar single-phase solid solution of five or more elements is supposedly rare and identifying the existence of such alloys has been challenging because of the vast chemical space of possible combinations. Herein, based on high-throughput density-functional theory calculations, we construct a chemical map of single-phase equimolar high-entropy alloys by investigating over 658,000 equimolar quinary alloys through a binary regular solid-solution model. We identify 30,201 potential single-phase equimolar alloys (5% of the possible combinations) forming mainly in body-centered cubic structures. We unveil the chemistries that are likely to form high-entropy alloys, and identify the complex interplay among mixing enthalpy, intermetallics formation, and melting point that drives the formation of these solid solutions. We demonstrate the power of our method by predicting the existence of two new high-entropy alloys, i.e. the body-centered cubic AlCoMnNiV and the face-centered cubic CoFeMnNiZn, which are successfully synthesized.

The field of metallurgy has been recently impacted by the emergence of high-entropy alloys (HEAs). In contrast to conventional alloys centering around one primary element with minor amounts of other elements, HEAs mix five or more elements at equal or near-equal compositions often in a single crystalline phase[1,2]. The seemingly surprising stabilization of multicomponent alloys against the formation of multiple phases and intermetallics (IMs) has been associated with the high configurational entropy[2] among other important factors[3]. HEAs can exhibit unusual properties[3,4] from exceptional toughness at cryogenic temperatures[5], to an outstanding combination of strength and ductility[6,7], high damage tolerance[8] and corrosion resistance[9]. While HEAs have first been mainly studied as structural materials, the field is now expanding to other areas such as electrocatalysis[10], thermoelectrics[11], and energy storage[12–17]. This is happening while the concept of high entropy stabilization is extended beyond metallic alloys with the development of high-entropy oxides and ceramics[18,19].

HEAs enjoy a vast compositional space. For equimolar quinary alloys, there are 658,008 candidates resulting from the combination of 40 elements. Yet, only a limited number of equimolar quinary single-phase HEAs have been observed experimentally over the last decade.

Computational approaches are called upon to understand the driving force towards the formation of HEAs and ultimately to accelerate the discovery of new HEAs with specific properties. Indeed, very limited regions in the compositional space have been explored and experimental screening alone would be formidable.

Numerous computational methods have been developed to predict the stability of single-phase solid solutions. Early models follow the Hume-Rothery theory[20,21] and rely on simple descriptors such as atomic radius mismatch and tabulated mixing enthalpy to induce the empirical rules for the formation of multicomponent solid solutions[22–25]. More sophisticated models additionally take into account the free energy of IM compounds[26–28], but are still oversimplified in that the IM phases are hypothetical and different definitions of the competing IM phases can lead to diverging predictions[25,29]. The CALPHAD (calculation of phase diagrams) method has been used to determine the phase formation of HEAs[30–32] although reliable thermodynamic databases are currently limited to a small number of elements[33]. The application of machine learning (ML) techniques to HEAs is also on the rise[34–41]. ML methods typically make use of the empirical descriptors already known to the existing single-phase solid

[1]Institute of Condensed Matter and Nanoscience (IMCN), UCLouvain, Chemin Etoiles 8, Louvain-la-Neuve 1348, Belgium. [2]Institute of Mechanics, Materials and Civil Engineering (iMMC), IMAP, UCLouvain, Place Sainte Barbe 2, Louvain-la-Neuve 1348, Belgium. [3]Univ. Bordeaux, CNRS, Bordeaux INP, ICMCB, UMR 5026, Pessac 33600, France. [4]Thayer School of Engineering, Dartmouth College, Thayer Drive 15, Hanover, NH 03755, USA. ✉e-mail: geoffroy.hautier@dartmouth.edu

solutions, and the relatively small training samples make extrapolating to less studied chemistry a bit hazardous.

First-principles methods offer unbiased insights into the thermodynamic properties of HEAs. These methods do not suffer from the fundamental issue with ML models when used for extrapolating to chemical regions that are not well experimentally explored. Enthalpies obtained from density functional theory (DFT) calculations have already been used in some semi-empirical models[26] and CALPHAD[33]. However, a full ab initio treatment of HEAs either involves supercells that are sufficiently large to accommodate the configurational disorder[42], or relies on statistical methods such as cluster expansions[16,43,44]. Either method poses a challenge due to the computational complexity and, when directly applied, is not suitable for high-throughput computational screening of HEAs.

In this work, we search possible single-phase HEAs among all equimolar quinary compositions from the combination of 40 metallic elements that are commonly used in alloys (Supplementary Fig. 1). This high-throughput computational screening is made possible by the use of a regular solution model[45,46] for which the interactions are described by binary terms and are obtained with DFT calculations. The thermodynamic stability of HEAs is determined by the Gibbs free energies of the system in solid solutions against those of the competing phases including IMs. Our computational model identifies 30,201 equimolar quinary HEAs, with the majority (75%) being BCC. Our work offers thus a map of the single-phase high entropy alloys indicating which chemistries favor the formation of these alloys. We identify that a high melting point ($T_m$) of the elements is among the most important driving factor in the formation of HEAs. In addition, some outlier elements, such as Al and Zn, are found to form HEAs easily despite their low melting point. We use our model to predict two equimolar single-phase HEAs, namely the BCC AlCoMnNiV and the FCC CoFeMnNiZn and we confirm experimentally their existence. The discovery of a BCC alloy and a FCC alloy analogous to the Cantor alloy (CoCrFeMnNi) but with the unusual element Zn is a compelling demonstration of how our thermodynamic model can suggest chemistries and new avenues to the development of HEAs.

## Results
### Computational model and validations

The thermodynamic stability of an alloy at a given temperature and pressure results from the competition between the Gibbs free energy of all competing phases. Here we use a regular solution model for all solid solution phases. The regular solution model combines an enthalpy model with a quadratic dependence in composition with an ideal configurational entropy (see Methods). We have previously shown that binary enthalpic interactions are sufficient to reproduce the mixing enthalpy of higher component (quaternary and quinary) random solid solutions[47]. Within this model, the Gibbs free energy of any random solid solution can be computed from a series of binary interactions that can be fitted, for instance on DFT. We have built such a database for a set of 40 elements using the special quasirandom structure (SQS) approach[48]. Using these regular solution Gibbs free energies, the competition between all phases can be assessed with the convex hull construction which directly compares the free energy of a

phase versus any linear combination of its subsystems. We additionally include competition from ordered, IM phases up to ternaries as provided by the AFLOW database[49]. We assume no configurational entropy for the IMs as they have well-defined occupancy of the lattice, thus bearing no configurational degree of freedom. More computational details are provided in the Methods section and our database of regular solution enthalpic parameters are available via an online repository[50]. The convex hull construction can be used to compute if an equimolar solid solution is stable for a given combination of elements. Our enthalpic model refrains from using any experimental parameters and is therefore fully ab initio. In addition to thermodynamic stability, our model informs the specific phase for stable HEAs or the decomposed phases for unstable ones.

The key parameter governing the phase stability assessment in the present study is the temperature $T$ at which the free energy is determined. In experiments, this temperature can be the synthesis or the annealing temperature. Our model can be used to predict if a given equimolar composition will form a single-phase solid solution or will decompose into several other phases. Naturally, higher temperatures favor the entropic contribution and stabilize the single-phase solid solution. We note that if an alloy can be made as a random single-phase solid solution at a high temperature, it will be likely to be retained when quenched. Prolonged annealing at intermediate temperatures may lead to phase decomposition, rendering the single-phase HEAs unstable as is the case with the Cantor alloy[1,51,52]. Nevertheless, the solid-solution phase formed at high temperatures can still be retained at room temperature following normal cooling rates[52]. So, the requirement to form a high-entropy single-phase equimolar solid solution will be here to show a (reasonable) temperature at which this single-phase is predicted to be stable according to our thermodynamic model.

To validate the predictive power of our model, we use 134 equimolar quaternary and quinary alloys that have been synthesized and structurally characterized experimentally[53–55]. This dataset includes 73 single-phase HEAs (Supplementary Table 2) and 61 multi-phase alloys (Supplementary Table 3). We use our Gibbs free energy model to see if a single-phase or multi-phase is predicted and if it agrees or disagrees with the experimental report. As the experimental data use different heat treatments, syntheses and annealing temperatures, we use a series of temperatures ($T$) from 800 to 1600 K in our model as typically used in the processing of metallic alloys. The predictive power of the model is assessed by the true positive rate (TPR) and the false positive rate (FPR), which are defined by the rate of predicted single-phase solid solutions from the 73 single-phase HEAs and from the 61 multi-phase alloys, respectively. A higher TPR indicates that the model is better at predicting true single-phase solid solutions, while the model with a higher FPR is considered to be overinclusive for single-phase solid solutions and is thus less reliable for predicting multi-phase alloys. The overall accuracy is determined by a combination of TPR and FPR as $[\text{TPR} \times 73 + (100 - \text{FPR}) \times 61]/134$. We find our model attaining a predictive accuracy of 74% at the optimal $T = 1350$ K (Table 1 and Supplementary Table 2). Specifically, our model predicts correctly 70% of the single-phase HEAs and 79% of the multi-phase alloys, suggesting that the model performs equally well regardless of the actual phase.

**Table 1 | Predictive metrics of the present thermodynamic model in comparison with various empirical rules (ERs) and free-energy models (FEMs)**

|  | Present ($T = 1350$ K) | ER1[22] | ER2[23] | ER3[24] | ER4[56] | FEM1[26] ($T = 1500$ K) | FEM2[27] ($T = 1350$ K) |
|---|---|---|---|---|---|---|---|
| TPR | 70 | 95 | 66 | 66 | 63 | 58 | 58 |
| FPR | 21 | 80 | 54 | 54 | 49 | 33 | 48 |
| Accuracy | 74 | 60 | 57 | 57 | 57 | 62 | 55 |

For the present and the two FEMs, temperature $T$ is chosen such that the best accuracy can be attained with the specific model.

To put our model in perspective, we apply four empirical rules (ERs)[22–24,56] and two free-energy models[26,27] developed previously to the same alloy dataset. In addition to the common criteria such as mixing enthalpy and entropy, the ERs rely mainly on atomic size mismatch whereas the two FEMs account for the formation of competing IMs (Supplementary Table 1). Notably, our model consistently outperforms all the ERs and the FEMs as shown in Table 1. Only the ER1[22] achieves a higher TPR, but this comes at the cost of a markedly high

FPR, showing that the model is strongly skewed towards the formation of single-phase solid solutions. The two FEMs are less predictive than our model irrespective of the temperature (Supplementary Table 2). The supremacy of our model is further made apparent by plotting the TPR vs FPR analogous to the receiver operating characteristic (ROC) analysis (Supplementary Fig. 2) as our model consistently provides better TPR and FPR than other models irrespective of the temperature used. It is noteworthy that Al-containing HEAs are normally rejected by

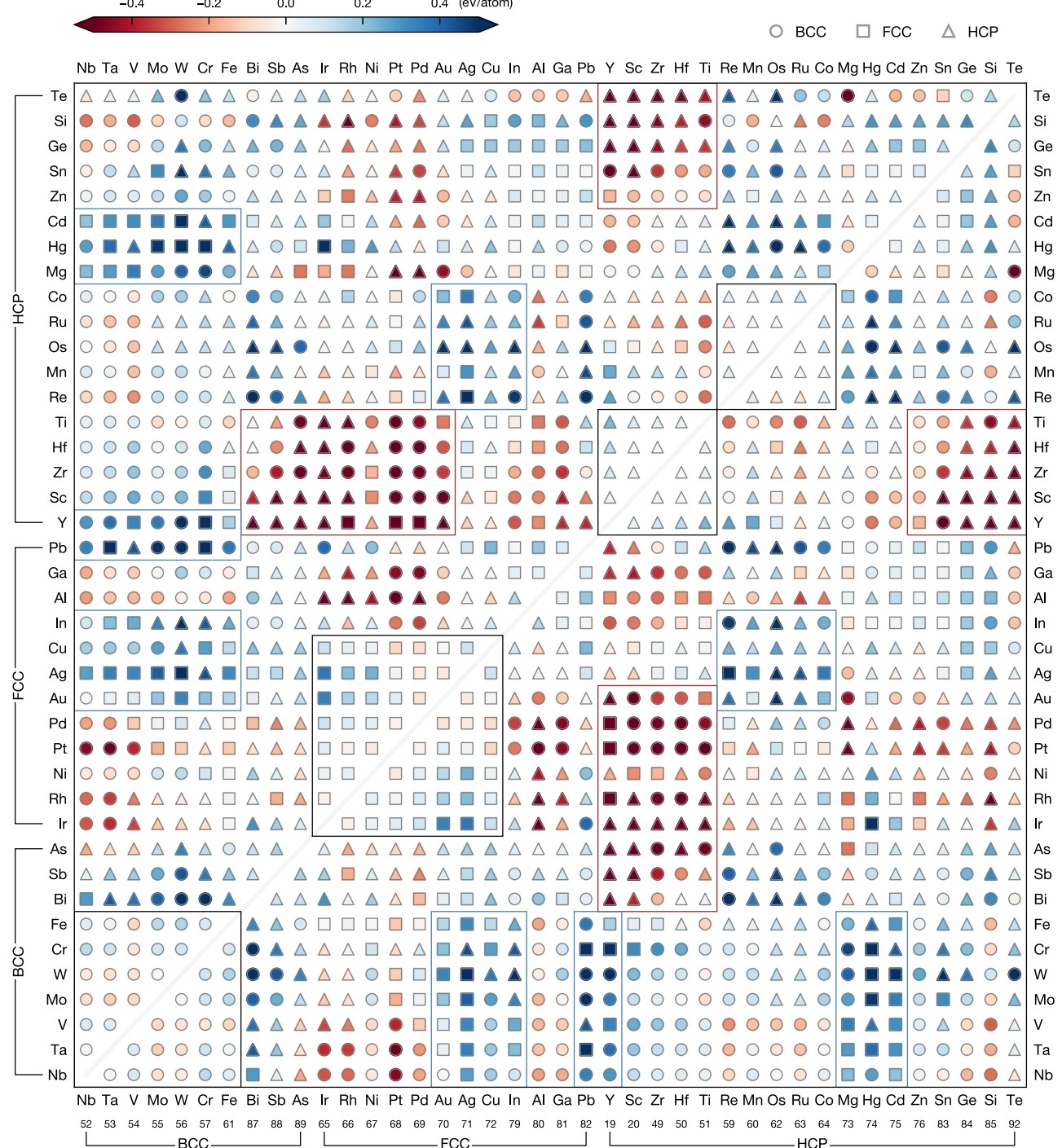

**Fig. 1 | Map of formation enthalpy for binary solid solutions, as represented by SQS, obtained from DFT calculations.** The formation enthalpy ($\Delta H^f$) is determined with respect to the ground-state elemental phases. The 40 elements are grouped by their lowest energy structure at 0 K (BCC, FCC, or HCP) and are sorted according to Pettifor's Mendeleev numbers[78]. Groups of elements mixing in the same crystal structure are shown by the black blocks. Groups of elements that strongly favor mixing ($\Delta H^f < -0.2$ eV/atom) are highlighted by the red blocks, whereas those strongly disfavoring mixing ($\Delta H^f > 0.2$ eV/atom) is highlighted by the blue blocks.

the ERs (except ER1) and FEMs mainly because the mixing enthalpies involving Al can be very negative and most models assume that solid solutions are unlikely to form if the mixing enthalpy is too strong. Nonetheless, such Al-containing HEAs are correctly predicted with our model by a large extent. A full breakdown of the results are given in Supplementary Table 3.

In addition to phase stability, a reasonable accuracy (74%) is achieved for predicting the structure of HEAs (Supplementary Table 4). This is comparable to the valence-electron concentration (VEC) model[57] although the original VEC model does not account for the HCP structure. If we make no distinction between FCC and HCP and treat the two simply as close-packed (CP), the accuracy is further improved to 84%. Therefore, our model is more capable of predicting if a HEA is BCC or CP (FCC or HCP) as we will discuss more later. In summary, our regular solution model is at least as effective as previous models and rely on a physically-driven approach based on DFT. It will therefore likely extrapolate better than approaches trained on a small dataset.

## Chemical map of high-entropy alloys

We now set out to navigate the huge chemical space of quinary alloys from the combination of 40 elements. To set ourselves in the best scenario for the formation of single-phase solid solutions, we choose here $T = 0.9T_m$ so that the entropic mixing contribution is maximized. As outlined before, we consider that HEA formed at high temperature can be quenched to room temperature preserving their single-phase nature. When tested with the same validation dataset, our model at $0.9T_m$ shows a high TPR of 84% but an FPR of 62%. The high TPR is more relevant in the current context of finding new HEAs.

Applying our model to the 658,008 possible equimolar quinary alloys, we find 30,201 potentially stable HEAs at $T = 0.9T_m$, which amounts to 4.6% of the quinary candidates. The majority of the stable HEAs (74%) are found in the BCC structures (Supplementary Table 5). Among the 7570 CP alloys, the model suggests a large amount of HCP alloys which disagrees with experimental knowledge[58]. As noted above, our model is less capable of discriminating between HCP and FCC in view of their small difference in energy (17 meV/atom when averaged over 75 known HEAs). Moreover, we tend to overestimate the stability of the HCP structures as the model does not take into account vibrational entropy which in general favors FCC vs HCP at high temperature. We estimate the effect of vibrational entropy using the CALPHAD entropy data for a set of 26 elements[32]. On average the vibrational entropy ($-S^{vib}$) of the HCP (BCC) structure is 24 (18) × $10^{-3}$ meV × $K^{-1}$ × atom$^{-1}$ higher than that of the FCC. This stabilization of the FCC vs HCP with temperature has been observed, for instance, in the Cantor alloy both experimentally[59,60] and computationally[61]. We note that the observed trend also applies to the phase stability analysis at lower temperatures albeit the predicted number of stable HEAs being reduced (Supplementary Table 5). The data on the thermodynamic stability of the 658,008 quinary alloys are accessible via an online repository[50].

One of the important factors driving the formation of quinary HEAs is the possibility for the five elements to enthalpically favorably mix in the solid solution. In our binary regular solid solution model, this is evaluated by the mixing enthalpy $\Delta H^{mix} = \frac{4}{25}\sum_{i,j>i}\Delta H_{i,j}^{mix}$ for a quinary equimolar alloy where $\Delta H_{i,j}^{mix}$ refers to the binary mixing enthalpy for the solid solution. The mixing enthalpy of a quinary solid solution is then the results of a sum over all mixing enthalpies of pair combinations of elements. For instance, the mixing enthalpy of the Cantor alloy (CoCrFeMnNi) is the result of the different combinations of binaries (Co−Cr, Co−Fe, Co−Mn, etc.). Figure 1 gives an overview of the tendency to mix for all pair combinations of the 40 elements. For each pair of elements, we plot the enthalpy of formation (i.e. with respect to the elemental phase in its lowest energy structure) and the crystal structure in the ground state at 0 K for the binary solid

solutions. This plot takes into account not only the mixing on a specific lattice (BCC, FCC, HCP) but also the competition between these lattices. Figure 1 gives a direct look at what pairs of elements will favor or disfavor mixing.

Groups of pairs of elements strongly favoring the formation of solid solutions have been indicated by red blocks in Fig. 1. For instance, FCC noble metals (Ir, Rh, Ni, Pt, Pd, Au) strongly mix with some HCP transition metals (Ti, Hf, Zr, Sc, Y), and the same set of HCP transition metals mix favorably with main-group elements (Te, Si, Ge, Sn, Zn). By contrast, we have also indicated with blue blocks highlighting the regions of disfavorable mixing. BCC refractory elements do not mix with a large group of CP elements (Cd, Hg, Mg, Y, Pb, In, Cu, Ag, Au). Among the other element pairs strongly disfavoring the mixing are some FCC elements (Au, Ag, Cu, In) with HCP elements (Co, Ru, Os, Mn, Re).

Our map can also be used to understand the prevalence of the stable HEAs in the BCC structure. More than 74% of the HEA form in the BCC structure while only 25% of the elements are BCC. While the crystal structure is likely to be maintained as a result of mixing two isostructural elements, it is not uncommon for certain elements to end up in solid solutions with a different structure than their elemental ones. Figure 2a indicates the statistics for the mixing of different structures in binary solid solutions. Remarkably, the BCC−FCC/HCP mixing leads to the majority of solid solutions being BCC, thereby explaining the large number of BCC HEAs. The elementwise analysis in Fig. 2b shows the preference for specific structures in binary solid solutions depending on the elements. The refractory elements (Nb, Ta, V, Mo, W) are remarkable for their strong tendency towards the formation of

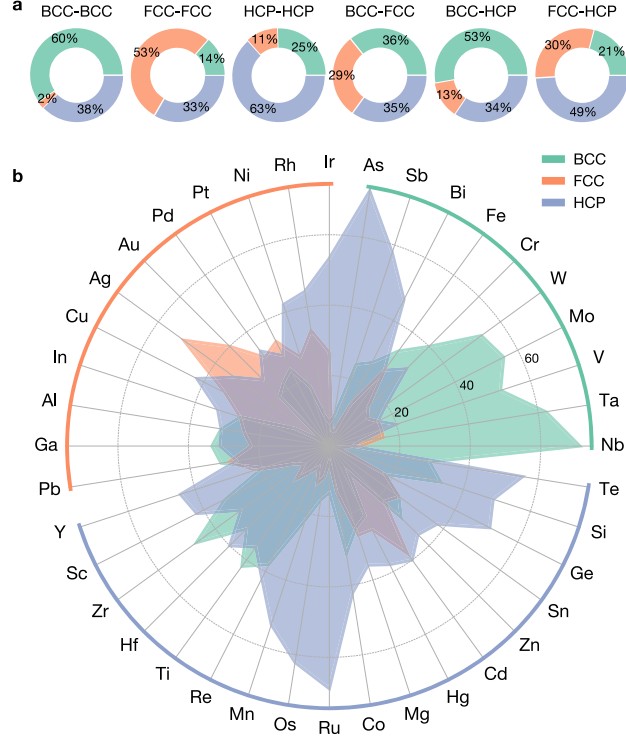

**Fig. 2 | Predicted structural preference of binary solid solutions. a** Structural preference of binary solid solutions formed through the mixing of two elements of specific ground-state structures ($S_1$−$S_2$) where $S_{1,2}$ refer to either BCC, FCC, or HCP. The statistics are based on the 40 candidate elements. **b** Ground-state structures of binary solid solutions summarized per constituent element. The values refer to the proportion of the structure among all structures found in the binary solid solutions containing the specific element.

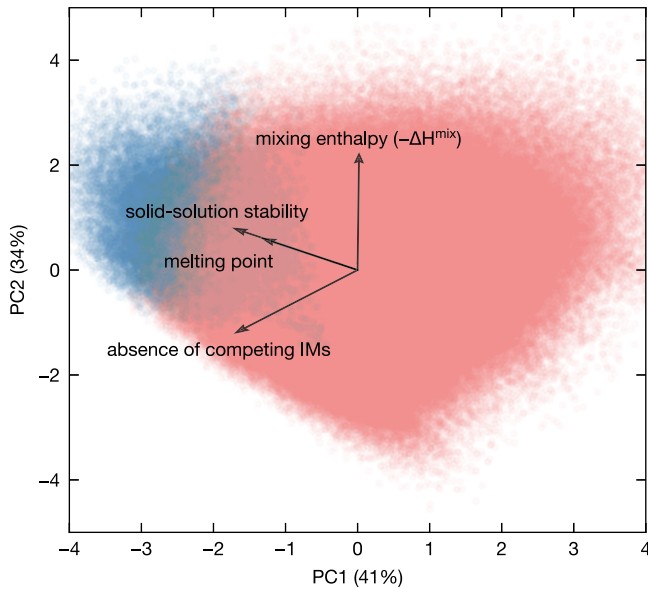

**Fig. 3 | Principal component analysis (PCA) biplot depicting the correlations between phase stability, melting point, mixing enthalpy, and competing IMs.** The degree of competing IMs is assessed by the change in the energy above the hull of the hypothetical HEA upon the introduction of IMs. The scores of the variables for the first two principal components (PC1 and PC2) are shown by the scattered points (blue for stable alloys and light red for unstable ones). The two components account for 75% of the explained variance ratio. The loadings are scaled by a factor of 2.5 for clarity.

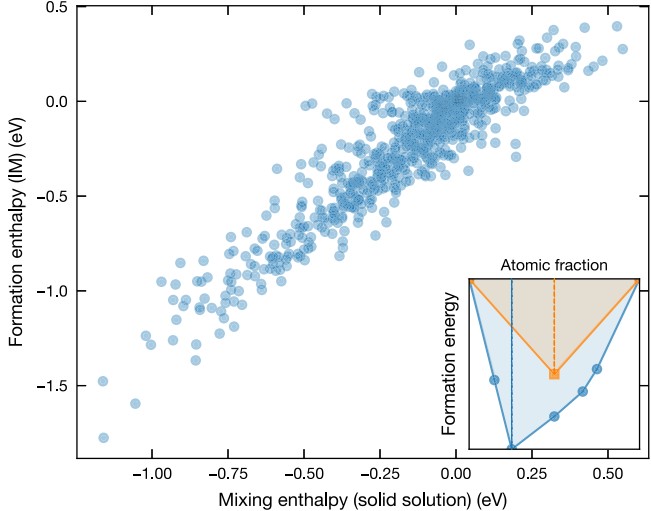

**Fig. 4 | Lowest formation enthalpy of IMs vs mixing enthalpy of equimolar solid solutions for binary systems.** The two quantities are illustrated by the schematic of the convex hull for solid solutions (orange) and IMs (blue) in the inset.

**BCC binary solid solutions.** These BCC refractory elements, when intermixed, consistently retain the BCC structure. FCC elements such as Al, Ga, or Pb favor as much BCC as CP structures despite their CP nature as elements. A similar effect is seen with HCP elements such as Zr, Hf, Ti, and Re. On the other hand, Mn, Os, and Ru are among the elements that are outstanding HCP formers, whereas Ag and Au are more likely to be found in FCC solid solutions.

Apart from mixing energetics, the formation of HEAs is also determined by melting point and IM formation. A high melting point will offer the possibility to entropically stabilize the solid solution through a high synthesis temperature. This is especially important when strong IM formation could destabilize the solid solution. To clarify the statistical importance of these three factors, we perform a principal component analysis (PCA) on the quinary HEAs dataset (Fig. 3). By projecting the multiple variables onto a lower dimensional space, the PCA is instructive in identifying the correlations among the variables. The two variables are more (anti)correlated if the loading vectors are more (anti)parallel, whereas they are uncorrelated if the loadings become orthogonal. The degree of stability is defined by the energy above the convex hull if the HEA is unstable or by the inverse energy above the hull (i.e. the equilibrium reaction energy) otherwise. The effect of IMs is quantified by the change in the energy above the hull upon the introduction of binary IMs. The stability of the alloys can be clearly discriminated from the PCA scores. It is apparent that the stability is mainly correlated with the melting point and, to a lesser extent, with the competing IMs and the mixing enthalpy of solid solutions. The formation of HEAs is therefore strongly driven by the possibility of a high synthesis temperature whereby the entropic stabilization effect is amplified.

The milder effect of the mixing enthalpy can be surprising at first but is rationalized by its correlation with IM formation. Elements that tend to mix strongly as a solid solution are also more likely to form ordered IM phases that in turn destabilize the solid solution. This was hinted at previously by Senkov and Miracle[27].

Taking all binary systems from the combination of the 40 elements, we explicitly show the linear correspondence between the formation enthalpy of IMs and the mixing enthalpy of solid solutions in Fig. 4. Following Fig. 1, we build a matrix indicating the competition between IMs and solid solutions for any given pair of the 40 elements (Supplementary Fig. 5). Interestingly, the Cantor alloy contains elements that tend to mix mildly together, not too favorably, not too disfavorably. While this could seem at first sight to be detrimental for HEA formation, this mild mixing in solid solution correlates with weak competition from IMs (Fig. 2 and Supplementary Fig. 6).

To probe how chemistry influences the stability of HEAs from the combination of melting temperature and mixing enthalpy, we summarize in Fig. 5 the elemental distribution of our predicted HEAs. We also add the elemental melting points. The prevalence of BCC HEAs is again remarkable. In addition to the mixing effect discussed previously where BCC is favored when mixing elements even if they alone form in CP structures, BCC is also favored in HEAs because a high melting point is often found among the BCC refractory metals. In fact, nearly 80% of the predicted quinary HEAs contain at least one refractory element (Cr, W, Mo, V, Ta, or Nb) (Supplementary Table 5), and 77% of these HEAs are stabilized in the BCC structure. Experimentally, a large number of refractory HEAs have been identified since the work of Senkov et al. in 2010[62] and they form the main body of single-phase HEAs as is clear from the collection of equimolar HEAs (Supplementary Table 3). The CP HEAs are largely formed by the noble FCC (Pd, Pt, Rh, Ir) and HCP elements (Re, Os, Ru), all of which have a relatively high melting point (1900–3400 K). Other noticeable CP HEA formers include Ni, Ti, Mn, and Co. While the formation of HEAs is overall favored by elements with a high melting point, we note two elements, namely Al and Zn, that are outliers to this melting-point rule. Despite the low melting point of Al (933 K) and Zn (692 K), the two elements can be found in a good amount of HEAs. We rationalize this as Al and Zn mix easily with elements of a higher melting point near 2000 K (Supplementary Fig. 7a). This is in line with the average melting point of the predicted HEAs containing Al (2100 K) and Zn (1900 K). An inverse trend is observed with the Cu, Ag, and Au, which form considerably fewer HEAs than the other elements with a comparable melting point (~1300 K). These group-11 elements are known to behave differently than many transition metals and their miscibility is low with elements of higher melting point (Supplementary Fig. 7b).

The analysis hitherto underscores the difficulty in finding HEAs that will be thermodynamically stable at low temperatures as the single-phase HEAs formed by entropic effect at high temperatures would instead be metastable, often destabilized by competition from IMs. Among the few alloys that our model predicts to be stable at $0.6T_m$ are a series of Sn and Cd alloys of low melting point (such as CdGaInMgSn, AgCdGaMgSn, and GaInMgPdSn), for which the formation is either driven by a weak mixing enthalpy of solid solutions in the absence of strong competition from IMs (CdGaInMgSn and AgCd-

GaMgSn) or stabilized by the favorable mixing enthalpy (e.g., Pd–$X$) despite the strong competition from IMs (GaInMgPdSn) (Supplementary Figs. 4 and 6).

## Discovery and synthesis of AlCoMnNiV and CoFeMnNiZn HEAs

HEAs containing Al are among the first synthesized HEAs[2] and exhibit intriguing phase-dependent strength-ductility properties[63–66]. According to our model, the AlCoMnNiV BCC solid solution shows an inverse energy above hull of − 0.3 meV/atom at 90% of the estimated melting point ($T_m$ = 1626 K) despite its strong mixing enthalpy of − 290 meV/atom, indicative of a subtle competition from the IMs. Some Al–$X$ pairs ($X$ = Co, Ni) indeed strongly favor the IM phase with respect to the solid solutions (Supplementary Fig. 5). The BCC solid solution of AlCoMnNiV is about 30 meV/atom more stable than the CP due to the presence of the three strong BCC formers (Al, Mn, and V). Given its moderate melting point and the strong competing IMs, the new BCC HEA AlCoMnNiV is an interesting test case for validating our model. As with many other Al-containing HEAs, most empirical models (except ER1) predict AlCoMnNiV unstable in a single phase. This composition has also to our knowledge never been reported in the experimental literature.

To experimentally validate our prediction, AlCoMnNiV is synthesized by arc-melting. Figure 6a shows the backscattered electrons (BSE) scanning electron microscopy (SEM) image of the microstructure of the alloy after heat treatment where single-phase, equiaxed grains are visible. The black spots are porosities due to the solidification. X-ray diffraction (XRD) analysis confirms that this HEA presents a BCC structure. Figure 6c shows the XRD pattern with the reflection associated to the BCC structure with a lattice parameter of 2.9 Å, highlighted in green, matching each peaks. The chemical homogeneity is confirmed by energy dispersive spectrometry (EDS) (Supplementary Fig. 8).

Compared to BCC HEAs, it is much more difficult to find HEAs stabilized in the FCC and HCP structures. Of all possible combination of quinary equimolar alloys, we predict that only 1% form CP alloys. CP HEAs exhibit some unique characteristics. For example, FCC HEAs are considerably more ductile than BCC ones, especially at low temperatures[67]. Bearing in mind the elemental cost, our

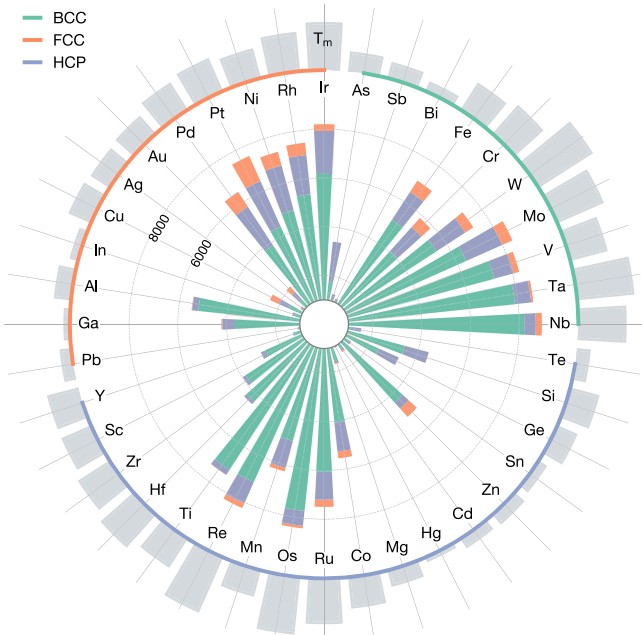

**Fig. 5 | Number of stable quinary solid solutions per constituent element.** The elements are grouped according to their ground-state structures. The elemental melting points are indicated by the height of the gray bars on the outskirt.

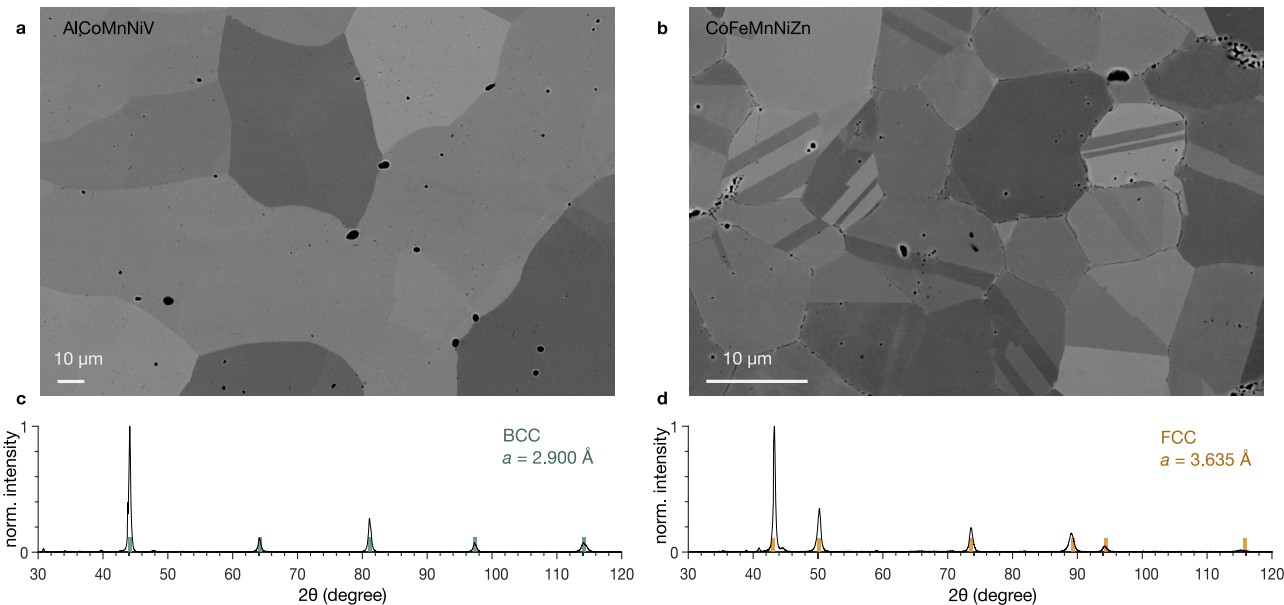

**Fig. 6 | Characterizations of the synthesized BCC AlCoMnNiV and FCC CoFeMnNiZn. a** and **b** Microstructures of the two HEAs characterized by (BSE) SEM. **c** and **d** XRD spectra of the two HEAs confirming the predicted structures. The vertical markers indicate the reflections for BCC and FCC structures.

model points to a series of cost-effective Zn-containing HEAs with a potential FCC structure. Here we choose the FCC CoFeMnNiZn as it is closely related to the Cantor alloy. While alloys with the same set of principal elements have been proposed[68,69], the phase of the equimolar CoFeMnNiZn has yet to be characterized. CoFeMnNiZn shows a weak mixing enthalpy of − 80 meV/atom and an inverse energy above hull of − 30 meV/atom at 90% of the melting point ($T_m$ = 1504 K). These values are in close agreement with those for the Cantor alloy, suggesting that the balance between the IMs and solid solutions is not disrupted by substituting Cr with Zn (Supplementary Fig. 6). Despite Zn being a low melting-point element, the absence of strong forming IMs leads to a high stability of the CoFeMnNiZn solid solution down to 900 K. In fact, our model indicates that Zn can be used to substitute any element of the Cantor alloy while still forming the FCC single-phase HEAs, although CoFeMnNiZn is predicted to be the most stable one. We note that Zn is not commonly used in HEAs but has been recently discussed in the field of biodegradable alloys[70].

We confirm experimentally the prediction of the model and synthesize a CoFeMnNiZn single-phase solid solution. Annealing twins, indicative of a FCC structure, are clearly visible in several grains in the microstructure shown in Fig. 6b. This is confirmed by the XRD pattern of CoFeMnNiZn in Fig. 6d which is indexed as FCC with a lattice parameter of 3.635 Å. The black spots observed in Fig. 6b corresponds to either MnO oxide particles or porosities (Supplementary Fig. 9). Mn is prone to oxidation whereas Zn has a low boiling point, limiting the processing routes available. It is worth mentioning that processing this alloy was a more arduous task than the previous one due to the sensitivity of Mn to oxidation and the low boiling point of Zn.

## Discussion

By accounting for the configurational entropy in addition to the mixing enthalpy, our ab initio-driven thermodynamic model achieves an accuracy up to 74% for predicting the phase stability of multi-component alloys, surpassing existing empirical rules and free-energy models. The predictive power of the model could be further improved by the inclusion of vibrational effects, magnetic ordering, and short-range ordering[25,71–73] that have been neglected in the present model. All of these effects would significantly increase the computational cost and prevent the large scale search we have reported on. Nevertheless, alloys suggested by our model could be used to perform a more refined modeling including these effects within a possible tiered screening approach for HEA discovery.

To search for potentially stable single-phase HEAs and to elucidate the mechanisms underlying their phase stability, we have navigated the vast chemical space of all quinary equimolar alloys from the combinations of 40 elements using our model. Among the 658,008 quinary alloys, we predict that 5% of them can be stabilized in a single-phase solid solution at near-melting point. The amount of predicted equimolar HEAs corresponds to the theoretical upper limit, and is significantly more than what has been reported in the literature. The predicted HEAs show a strong tendency to form BCC phases, in line with the large body of BCC HEAs that have already been identified. Our model suggests that the prevailing BCC phase originates from a combined effect of the high melting of the constituent BCC elements, which are often refractory, and a favorable mixing of elements on a BCC lattice. The high melting point is in fact one of the main driving forces for the single-phase HEA formation. By that token, many closed-packed alloys, such as the FCC Cantor alloy, are more the exception than the rule as they normally contain zero to very few refractory elements.

The map of binary interactions presented here is instructive in rationalizing and predicting the chemistries that are likely to lead to new HEAs. The series of Al- and Zn-containing HEAs show that non-refractory and cost-effective HEAs with a relatively low melting point

can be stabilized by the subtle enthalpic competition between IMs and solid solutions. The successful synthesis of the BCC AlCoMnNiV and the FCC CoFeMnNiZn signals the promising application of our current approach towards the quest for new HEAs.

While our work does not inform properties other than phase stability, additional computational screenings driven by specific desired properties can be envisaged in combination with our thermodynamic model. In addition, the present approach is applicable to a range of technologically relevant temperatures and can readily be applied to alloys deviating from the equimolar composition.

## Methods

### Binary solid-solution model

The Gibbs free energy of mixing at temperature $T$ can be expressed as $\Delta G^{mix} = \Delta H^{mix} - T\Delta S^{mix}$. Within the binary regular solution model, the enthalpy of mixing $\Delta H_{mix}$ of an $n$-component system can be written as a linear combination of the pair interactions among the constituent elements

$$\Delta H^{mix} = \sum_i \sum_{j>i} \Omega_{ij} c_i c_j, \tag{1}$$

where $\Omega_{ij}$ are the binary interaction between atoms $i$ and $j$ and the sum runs through all combination of pairs, and $c_i$ is the molar fraction of the $i$th element. The mixing entropy in a regular solution is the ideal mixing entropy

$$\Delta S^{mix}_{ideal} = - R \sum_i c_i \ln c_i, \tag{2}$$

where $R$ is the gas constant. The binary interaction is obtained from the enthalpy of mixing of the binary system as

$$\Omega_{ij} = 4\Delta H^{mix}_{ij} = 4\left[E^{SQS}_{ij} - \frac{1}{2}(E_i + E_j)\right], \tag{3}$$

where $E^{SQS}_{ij}$ is the total energy of the binary system represented by the special quasirandom structure (SQS), and $E_i$ is the total energy of the elemental system $i$ in the same lattice as the parent binary structure. The 16-atom SQS structures for the FCC, BCC, and HCP structures used in this work are generated by the ATAT suite of software[74], whereby pair (triplet) interaction up to the 6th (3rd) nearest neighbor are taken into account. This Gibbs free energy of mixing is defined for a given structural lattice and computed on FCC, HCP, and BCC lattice.

For the construction of an energy convex hull, the Gibbs free energy is used and can be easily obtained from $\Delta G^{mix}$

$$\Delta G = \Delta G^{mix} + \sum_i c_i E_i. \tag{4}$$

The convex hull analysis is carried out using pymatgen[75].

### Competing intermetallics

For a given quinary alloy, we search all its binary and ternary compounds using the AFLOW ICSD and LIB collections of IMs. We construct a convex hull of formation energy for the available IMs for a specific binary or ternary composition, from which the stable compounds and the unstable compounds with an energy above the hull of less than 10 meV/atom are chosen for a refining DFT computation using the same parameters as for the solid solutions. The 10 meV/atom cutoff threshold takes into account the uncertainties arising from the differences between the computation parameters (e.g., pseudopotentials, kinetic energy cutoffs, and k-point samplings) used by the AFLOW dataset and our present calculations. Supplementary Fig. 3 shows that the threshold of 10 meV/atom suffices to include as many

stable IMs as the numerical uncertainty of the formation energy is indeed typically within this value. The calculated IM entries (about 9100 binaries and 7800 ternaries) are then added together with the solid solutions to the energy convex hull in order to assess the competition from the IMs.

While we consider the IMs up to the ternaries, the effect of competing phases is largely captured by the binary systems. If only the binary IMs are considered in our model, the number of stable quinary single-phase HEAs would be 35,608, i.e. about 10% more than in the presence of ternary IMs. Any higher-order IMs would have a negligible effect.

## Density-functional theory calculation

DFT calculations for the SQSs and IMs are performed within the Perdew-Burke-Ernzerhof (PBE) generalized-gradient approximation using the VASP code[76,77]. The planewave energy cutoff is 500 eV, and a grid density of 2000 **k** points per number of atoms is used for sampling the Brillouin zone. The atomic positions are relaxed until the forces are smaller than 0.02 eV/Å. For the BCC and FCC SQSs, the lattice parameter is optimized while the cell shape is intact. For the HCP SQS, an additional constraint is imposed on the $c/a$ ratio. Specifically, we fully optimize the HCP structure for each of the constituent elements and take the average $c/a$ value for the HCP SQS. All calculations are spin-polarized and initialized with a ferromagnetic configuration.

## Processing and analysis of AlCoMnNiV and CoFeMnNiZn

AlCoMnNiV is prepared by arc melting of pure elements ( > 99.9% purity). A pre-alloy of manganese and nickel is made to avoid Mn evaporation issues. Furthermore, the Mn chips are deoxidized prior to casting using 50% HCl solution. The alloy is re-melted at least five times in the arc furnace before being heat treated at 1373 K for 24 h.

The processing of CoFeMnNiZn is more challenging due to the presence of Zn. Due to its low melting (693 K) and boiling point (1180 K), arc or induction melting is not appropriate. Instead, CoFeMnNiZn is obtained by mechanical alloying of pure elements (powders of > 99.9% purity). Powder pre-alloying is carried out before sintering for densification. 10-mm-thick pellets are compacted in a die of 12 mm in diameter with of force of 50 kN and heat treated in Ar-filled quartz capsule for 5 days at 1073 K followed by 5 hours at 1273 K. The heat-treated pellets are then ground and the resulting powder is densified by spark plasma sintering (SPS). The sintering is carried in a 30-mm die at 48 MPa and 1123 K with a holding time of 2.5 min. The heating cycle is carried out under Ar atmosphere.

Sample preparations follow standard practices starting with mechanical polishing with SiC, followed by 6 and 1 $\mu$m diamond paste polishing. The final step consists in polishing with a solution containing silica oxide particles in suspension (OPS). The microstructure is characterized by scanning electron microscopy (SEM) at 15 keV. Local chemical analysis is measured by energy-dispersive X-ray spectrometry (EDS). The x-ray diffraction (XRD) with Cu radiation (wavelength of 1.541 Å) operated at 30 kV and 30 mA is performed on unetched samples to characterize the crystallography.

## Data availability

The phase stability analysis of the 658,008 quinary alloys is available as a csv file at https://doi.org/10.5281/zenodo.7633180. The binary interactions for the 40 elements and the formation energies of the binary and ternary intermetallics are also available as json files and can be downloaded from the repository.

## Code availability

The code used in the present work for the phase stability analysis is available at https://doi.org/10.5281/zenodo.7633180.

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

## Acknowledgements

The research was funded by the Walloon Region under agreement No. 1610154-EntroTough in the context of the 2016 Wallinnov call. Computational resources were provided by the Consortium des Équipements de Calcul Intensif (CÉCI), funded by the Fonds de la Recherche Scientifique de Belgique (F.R.S.-FNRS) under Grant No. 2.5020.11 and by the Walloon Region. The present research benefited from computational resources made available on the Tier-1 supercomputer of the Fédération Wallonie-Bruxelles, infrastructure funded by the Walloon Region under grant agreement No. 117545. G.H. acknowledges support as well from the U.S. Department of Energy, Office of Science, Office of Basic Energy Sciences Established Program to Stimulate Competitive Research (EPSCoR) program under Award Number DE-SC-0021347.

## Author contributions

W.C. performed formal analysis and investigation, developed methodology, and wrote the manuscript. A.H. conducted experimental syntheses and characterizations and co-wrote the manuscript. G.B. was involved in data curation and methodology development. S.G. contributed to the investigation, provided resources, and edited the manuscript. P.J. and G.H. conceptualized the project, acquired funding, administered the project, and edited the manuscript.

## Competing interests

The authors declare no competing interests.
