## [Peer Review File · Nature Communications]

A map of single-phase high-entropy alloysReviewers' Comments:

Reviewer #1:

Remarks to the Author:

In this paper the authors describe a screening of equimolar high entropy alloys using a solid solution model based on binary first principles data. The authors also provide experimental results for some of the predicted structures. This work provides valuable design principles for the exploration of multicomponent concentrated solid solution alloys.

Given the combinatorial complexity of the problem, first principles calculations for even a fraction of the possible alloys while considering multiple crystal structures and intermetallic compounds would be prohibitively expensive. Thus, this paper can guide researchers to promising high entropy alloys.

While the approach presented here appears to be very valuable, the authors should discuss the possible limitations of their method. In particular, the authors chose to consider materials where the energies are within 10meV above the hull. It is obvious that excessively large favoring of intermetallics would be detrimental to the formation of solid solutions, so qualitatively this choice makes sense, but what is the quantitative argument for the specific value? Also, as the model is only based on data from binary materials (and their previous Scr. Mater. paper discussed this as well, justifying this model by comparison with multicomponent calculations) to what extent could possible multicomponent intermetallics play a role in destabilizing the single solid solution phase? Magnetism might play an important role in some materials (most likely only at lower temperatures) as it can influence the electronic structure for materials that are close to the transition between a magnetically and non-magnetically ordered state.

Finally, it would be nice to see a comparison of the method in this paper with experimentally known HEAs, both correct / false prediction of formation / non-formation of HEAs.

In conclusion, I see this paper as a very valuable addition to the study of the formation and design of solid solution single phase high entropy alloys and I would recommend it for publication if a critical assessment of the limitation and validity of the approach are included.

Reviewer #2:

Remarks to the Author:

The submitted manuscript "A map of single-phase high-entropy alloys" used a well-established binary regular solid solution model and DFT calculations to study the phase stability of equiatomic five-element high entropy alloys, and the outcome of a single phase was specifically targeted. The physics-based model has managed to achieve a reasonable prediction accuracy compared to empirical models, and the high throughput method has generated important statistical information (e.g., correlation between elements and tendency to form a specific phase/intermetallic, correlation between melting points and phase stability) to guide alloy design in the high entropy alloy community. However, listed below are certain critical points that detract the overall quality and the potential impact of the paper. A major revision is required for this manuscript to be considered for publication.

Major drawbacks:

1. The main screening targets of this manuscript are stable single-phase high entropy alloys that do not involve any prediction on their properties, which is the most important aspect of alloy design. The reader is easily overwhelmed by the sheer number of stable single-phase high entropy alloys (over 30,000) yet still confused by which one among them can meet the desired design criteria. In addition, despite the simplicity of their microstructure, equiatomic, single-phase high entropy alloys often do not possess superior or comparable mechanical properties to their non-equiatomic, multi-phase counterparts. For example, current challenges of single-phase FCC HEAs include lack of strength at

room and higher temperatures and those of single-phase BCC HEAs include lack of room and high temperature tensile ductility. In both cases, deviation from equiatomic composition and single-phase microstructure is required to improve the mechanical properties of these materials. Recent successful high-throughput screening studies published in comparable journals (e.g., <https://doi.org/10.1038/s41467-021-24523-9>, <https://www.science.org/doi/10.1126/sciadv.abo7333>) include clear design criteria (i.e., desired properties and microstructure at specific service temperatures) and experimental verification of those properties (e.g., hardness and ductility). The lack of the above aspects is likely to significantly lower the impact of the current manuscript.

2. Only considering the phase stability at a synthesis temperature of $0.9 T_m$ and assuming the retainment of single phase at room temperature is significantly flawed. For example, the target service temperatures of BCC refractory high entropy alloys are $\sim 1400^\circ\text{C}$ for turbine blades and $\sim 2000^\circ\text{C}$ for hypersonic vehicle applications, which are close to $0.5T_m$ and $0.75T_m$ of these alloys, respectively. The phase stability at these service temperatures is of critical importance and is completely missed by the current paper.

3. On a practical note, despite maintaining a single phase, the combination of some HEAs with elements of very different melting points, vapor pressure, and oxidation resistance makes fabrication very difficult. This is pointed out by the authors themselves on the processing of the CoFeMnNiZn alloy. Furthermore, other aspects such as the cost or toxicity of a lot of metallic elements can also make the practical use of the alloys prohibitive.

The reviewer suggests the authors include further screening criteria (e.g., mechanical properties and microstructure at target service temperatures) and verify them experimentally. In many cases, the criteria of single phase and equiatomic composition can be relaxed.

Minor issues:

1. The introduction needs some rephrasing in that several phrases and arguments are inappropriate: Line 28 "disrupted"; Line 32 high configuration entropy does not seem to be the main reason for stable single phases (see <https://doi.org/10.1016/j.actamat.2016.08.081>); Line 33 only a very limited number of HEAs has shown promising properties; Line 73 limiting the interaction to binary terms needs to be justified. A discussion is needed in the main text or the supplementary material.

2. A brief discussion of the possible effect of chemical short-range order (CSRO) is needed. A recent kinetic Monte-Carlo simulation (<https://doi.org/10.1016/j.commat.2021.110670>) shows that the time scale of forming atomic CSRO is on the order of microseconds to milliseconds, which means that CSRO can form during quenching.

3. The usefulness of the statistical analysis given in figures 3 and 4 is limited because they give little information in terms of the design of a specific HEA. It is advised that the authors put them into the supplementary material.

4. The paragraph from Line 217-228 gives little information and implies strongly that the understanding of quinary HEAs relies on case-by-case scrutiny, which does not help the idea of high throughput screening.

5. The authors need to describe how representative figures 6 and 7 are (maybe they belong to a small equiaxed grain region surrounded by columnar grains with large segregation) and provide the quantitative EDS results.

Reviewer #3:

Remarks to the Author:

The paper, "A map of single-phase high-entropy alloys", describes a new model, developed using DFT databases, to predict whether high-entropy alloys (HEAs) across a wide range of compositions will be stable. The paper also describe the successful experimental synthesis of two alloys predicted by the model to be stable. Overall, the paper is well written and describes a valuable contribution to the study of high-entropy alloys (HEAs). The paper describe the model clearly and includes a useful discussion of various results from the model. I recommend the paper for publication, and provide the following comments that may help the authors clarify some of the presentation of results and provide suggestions for some additional discussion:

Comments:

- In the introductory paragraph the authors include the sentence "The seemingly surprising stabilization of HEAs against 31 the formation of multiple phases and intermetallics has been credited to the high configurational entropy of multicomponent solid solutions." Ref. 3 cited here (D. Miracle and O. Senkov, A critical review of high entropy alloys and related concepts, Acta Mater. 122, 448 (2017).), and the authors previous work in Ref. 48 (G. Bokas, W. Chen, A. Hilhorst, P. Jacques, S. Gorsse, and G. Hautier, Unveiling the thermo- dynamic driving forces for high entropy alloys formation through big data ab initio analysis, Scr. Mater. 202, 114000 (2021).) makes clear that the role of entropy is important, but that it is hard to clearly distinguish, especially across HEAs in general, in a way this sentence does not seem to capture. While that complexity cannot be fully described in this introduction, it would be useful to at least hint at it.
- On pg. 5 the authors state "We assume no configurational entropy for these ordered phases." It seems like the effect of this approximation should be discussed more carefully in HEAs. See for example Ref. 3, sec. 2.2.2.
- Particularly due to the Methods section being located at the end of the paper, I found that on first reading I had some trouble understanding what exactly various quantities in the figures referred to, whether because the calculation method was unclear or whether I was unsure if the data was experimental vs model predictions. Some examples:
 - In Fig. 1, "...formation energy for binary solid solutions obtained from DFT..." could be changed to "...formation energy for binary solid solutions, as represented by SQS, obtained from DFT..."
 - In Fig. 2a, "Structural preference of binary solid solutions..." could be changed to "Predicted structural preference of binary solid solutions ...", and in Fig. 2b, "Ground-state structures of binary solid solutions..." could be changed to "Predicted ground-state structures of binary solid solutions..."
 - Fig. 4. is a bit confusing. Using the singular "Enthalpy of formation (intermetallics)" instead of "Lowest intermetallic formation energy", or something similar, seems unnecessarily confusing. It also seems confusing to compare "Enthalpy of mixing (solid solution)" to the inset orange line with a single point from the SQS calculations instead of perhaps the curve calculated from the regular solution model. I understand the equivalence after reading through the entire paper and methods section, but it was not immediately clear when first viewing the figure.
- On page 13, line 259, "Among the few alloys that are stable at low temperatures are a series of Bi and Ag alloys of low melting point" -> "Among the few alloys that our model predicts to be stable below X% Tm are a series of Bi and Ag alloys..."
- FIG. S2. "Enthalpy of formation of binary solid solutions" -> "Enthalpy of formation of binary solid solutions, obtained from DFT calculations of SQS, for ..." or "Predicted enthalpy of formation of binary solid solutions for ..."
- FIG. S3 and S5: Similar to S2.
- Could the authors clarify the criteria for "strongly favoring" or "strongly disfavoring" mixing used to draw the lines in Fig. 1?
- Could the authors discuss the effect, if any, of validating the model on the prediction of experimentally found stable compositions but not on known unstable compositions?
- Did the authors attempt to synthesize any other compositions that did not form stable solid solutions?
- The ER models have more false predictions of unstable compositions in the set of 75 stable compositions than the authors model. Do the ER models also have fewer false predictions of stable compositions?

Places where small edits are necessary:

pg. 4, line 80: "is a very important driving factor the formation of HEAs but identify..." -> "is a very important factor driving the formation of HEAs, but identify..."

pg. 5, line 96: "can be fitted for instance on DFT." -> "can be fitted, for instance on DFT."

pg. 12, line 232: "In addition, to the mixing effect discussed previously where..." -> "In addition to the mixing effect discussed previously, where..."

pg. 13, line 253: "The analyses hitherto are based on the hypothesis that the synthesis temperature can be as high as near melting point and that HEA alloys will be quenched to their operation temperature (e.g., room temperature)." -> "The analyses hitherto assume that the synthesis temperature can be as high as 90% T_m and that..."

Response to Reviewer #1

In this paper the authors describe a screening of equimolar high entropy alloys using a solid solution model based on binary first principles data. The authors also provide experimental results for some of the predicted structures. This work provides valuable design principles for the exploration of multicomponent concentrated solid solution alloys.

Given the combinatorial complexity of the problem, first principles calculations for even a fraction of the possible alloys while considering multiple crystal structures and intermetallic compounds would be prohibitively expensive. Thus, this paper can guide researchers to promising high entropy alloys.

We thank the reviewer for the positive recommendation.

While the approach presented here appears to be very valuable, the authors should discuss the possible limitations of their method. In particular, the authors chose to consider materials where the energies are within 10 meV above the hull. It is obvious that excessively large favoring of intermetallics would be detrimental to the formation of solid solutions, so qualitatively this choice makes sense, but what is the quantitative argument for the specific value?

There might have been a confusion here in the reason for the 10 meV/atom criteria for intermetallics chosen from the AFLOW database. We are not using exactly the same parameters (k-point density, pseudopotential, etc...) than AFLOW so we have to recompute the energies with our parameters. We do not expect the difference between our computed energies and the one from AFLOW to differ dramatically though. Hence, our use of a (very loose) criteria of 10 meV/at is to make sure we include enough phases that could be stabilized on the hull with our parameters. We have rephrased this part of the manuscript to be clearer on the reason and process involved. To justify this 10 meV/at threshold, we provide in Fig. S3 of the revised SI a comparison of the formation energies of the 20 binary intermetallic subsystems pertinent to the Cantor alloy CoCrFeMnNi and the refractory alloy MoNbTaVW. Generally the two convex hulls (one from our calculated formation energies and the other from the AFLOW data) agree very well. In some cases, the unstable intermetallics in the AFLOW dataset become stable in our calculations (see e.g., CoMn, MoNb, MoW, NbTa) and the energies above hull for these systems are well within

10 meV/atom. The threshold of 10 meV/atom therefore suffices to include as many stable intermetallics since the numerical uncertainty of the calculated formation energy is typically within this value.

Also, as the model is only based on data from binary materials (and their previous Scr. Mater. paper discussed this as well, justifying this model by comparison with multicomponent calculations) to what extent could possible multicomponent intermetallics play a role in destabilizing the single solid solution phase?

Following the reviewer's comment, we incorporate the ternary intermetallics from the AFLOW LIB3 dataset. Upon the inclusion of the ternaries, the number of stable quinary single-phase HEAs reduces to 30,201, which is 10% fewer than previously reported with only the binary competing intermetallics. We thus conclude that including any higher component intermetallics would have no major effect.

In the revised manuscript, we have updated the results to account for the effect of the ternary intermetallics. Except for the slightly reduced number of stable HEAs, the conclusions remain largely intact.

Magnetism might play an important role in some materials (most likely only at lower temperatures) as it can influence the electronic structure for materials that are close to the transition between a magnetically and non-magnetically ordered state.

Indeed, HEAs can exhibit subtle magnetic interactions, which have been shown to correlate with the short-range order and subsequently affect the formation energy and the phase stability. We have performed all computations with spin polarization but initializing the magnetic moments as ferromagnetic. Hence, we do not model subtle magnetic ordering effects. Adding those effects would likely improve the model but is beyond the current high-throughput capabilities. This is a limitation of the present model but might be less relevant at the high temperature used for synthesis as rightfully suggested by the reviewer. We have pointed it out in the discussion section of the revised manuscript.

Finally, it would be nice to see a comparison of the method in this paper with experimentally known HEAs, both correct / false prediction of formation / non-formation of HEAs.

In addition to the 73 single-phase HEAs originally considered in the manuscript, we further include 61 experimentally confirmed multi-phase alloys to the test dataset. Furthermore, instead of 90% of melting point, we consider temperatures ranging from 800 to 1600 K which are the typical annealing temperatures for HEA processing. This is important when comparing to previous experimental data. This results in a more realistic and balanced model validation. We would like to mention that our model correctly predicts 70% of the single-phase HEAs, and 80% of the multi-phase alloys that tend not to form HEAs, consistently outperforming the other empirical rules and free-energy models irrespective of the phase of the alloys.

We have revised the manuscript to adopt the changes in the model validations. Tables 1, 2, and S2–4 have been revised accordingly.

In conclusion, I see this paper as a very valuable addition to the study of the formation and design of solid solution single phase high entropy alloys and I would recommend it for publication if a critical assessment of the limitation and validity of the approach are included.

We once again thank the reviewer for the recommendation.

Response to Reviewer #2

The submitted manuscript “A map of single-phase high-entropy alloys” used a well-established binary regular solid solution model and DFT calculations to study the phase stability of equiatomic five-element high entropy alloys, and the outcome of a single phase was specifically targeted. The physics-based model has managed to achieve a reasonable prediction accuracy compared to empirical models, and the high throughput method has generated important statistical information (e.g., correlation between elements and tendency to form a specific phase/intermetallic, correlation between melting points and phase stability) to guide alloy design in the high entropy alloy community. However, listed below are certain critical points that detract the overall quality and the potential impact of the paper. A major revision is required for this manuscript to be considered for publication.

We are glad that the reviewer recognizes the importance of the statistical informa-

tion on the HEA phase stability generated by our work. In what follows we address the critical points raised by the reviewer.

Major drawbacks:

1. *The main screening targets of this manuscript are stable single-phase high entropy alloys that do not involve any prediction on their properties, which is the most important aspect of alloy design. The reader is easily overwhelmed by the sheer number of stable single-phase high entropy alloys (over 30,000) yet still confused by which one among them can meet the desired design criteria. In addition, despite the simplicity of their microstructure, equiatomic, single-phase high entropy alloys often do not possess superior or comparable mechanical properties to their non-equiatomic, multi-phase counterparts. For example, current challenges of single-phase FCC HEAs include lack of strength at room and higher temperatures and those of single-phase BCC HEAs include lack of room and high temperature tensile ductility. In both cases, deviation from equiatomic composition and single-phase microstructure is required to improve the mechanical properties of these materials. Recent successful high-throughput screening studies published in comparable journals (e.g., <https://doi.org/10.1038/s41467-021-24523-9>, <https://www.science.org/doi/10.1126/sciadv.abo7333>) include clear design criteria (i.e., desired properties and microstructure at specific service temperatures) and experimental verification of those properties (e.g., hardness and ductility). The lack of the above aspects is likely to significantly lower the impact of the current manuscript.*

The two studies highlighted (<https://doi.org/10.1038/s41467-021-24523-9> and <https://www.science.org/doi/10.1126/sciadv.abo7333>) by the reviewer report on a search for new alloys combining phase stability and properties while our work only focuses on phase stability. So, at first sight it might look like we are doing “less”. However, these two papers are quite different from our paper in scope as they focus on a given set of elements (Al-Cr-Fe-Mn-Ti or Co-Cr-Fe-Mn-Ni) and explore the composition space to optimize certain properties. Our approach is to look among all the possible combinations of elements to search for new equimolar single-phase high entropy alloys. So, we are exploring a much larger space of possible elements to combine. The two approaches can be complementary though and we can imagine some combinations of elements being suggested by our model to motivate a more deep-dive studies optimizing the compositional space within these elements and including properties. However, it is not in our opinion realistic to expect such deep-dive study to be performed in this paper. The paper is already

dense in terms of results.

More fundamentally, we would like also to emphasize that high entropy alloys are of interest in many fields and not only in mechanical alloys (which is itself a vast field with many properties to explore). High-entropy alloys are for instance a growing sub-field in electrocatalysis or thermoelectrics. A common need to all these fields is to predict and understand phase stability. We have kept an application agnostic approach in this paper focusing on the phase stability issue common to all these fields. We expect each community to use our results to launch further theoretical and experimental studies in new and unexpected directions (this could include but will be not limited to hardness and ductility optimization as suggested by the reviewer).

We respectfully disagree with the reviewer when it is suggested that focusing “only” on phase stability is not impactful. Understanding phase stability of materials is a long enduring goal of materials science and inorganic chemistry. Textbook rules and core knowledge in these fields focus on phase stability: the Hume-Rothery rules for alloys, the Pauling rules in inorganic chemistry, the Goldschmidt factor for perovskites to cite a few fundamental rules, all “only” focus on stability. The famous CALPHAD (CALculation of PHase Diagrams) method, first developed in the 1970s by a group of researchers led by Larry Kaufman, is widely used in materials science and engineering to predict “only” phase stability (when thermodynamic databases are available) because it is the cornerstone of the design of a wide range of materials. As far as the scientific literature is concerned, we can cite a few recent studies focusing as well “only” on the understanding and prediction of phase stability of inorganic compounds, perovskites or ternary nitrides: <https://doi.org/10.1126/sciadv.aay5606>, <https://doi.org/10.1038/s41563-019-0396-2>, <https://www.pnas.org/doi/10.1073/pnas.1719179115>. These studies have been published in journals of similar impact than Nature Communications (Sci. Adv. Nat. Materials, PNAS) and none of these studies target a specific application or perform an application-oriented design.

As for the note from the reviewer on the interest of equimolar single-phase alloy, this is a still a debated question in the literature (along the interest for high-entropy alloys versus traditional alloys in general) that is strongly linked again with the considered application. We are hopeful that our work will help identify more single phase quinary alloys (a quite rare occurrence) and thus build a larger body of

experimental and theoretical data to answer the fundamental question of the true and general effect of quinary solid solutions.

2. Only considering the phase stability at a synthesis temperature of $0.9T_m$ and assuming the retainment of single phase at room temperature is significantly flawed. For example, the target service temperatures of BCC refractory high entropy alloys are $\sim 1400^\circ\text{C}$ for turbine blades and $\sim 2000^\circ\text{C}$ for hypersonic vehicle applications, which are close to $0.5T_m$ and $0.75T_m$ of these alloys, respectively. The phase stability at these service temperatures is of critical importance and is completely missed by the current paper.

The temperature of $0.9T_m$ is chosen to maximize the probability of finding new HEAs. It is somewhat of a best-case scenario where the synthesis temperature is pushed to the highest to favor solid solution mixing. The reviewer's comment is relevant for high temperature alloys but less important for other important field where high-entropy alloys are of interest for instance cryogenic or room temperature structural alloys or electrocatalysis where temperatures of operation are much lower. We insist that we are trying to offer a general view of high- entropy alloy stability without delving into the needs of specific fields and community. The community interested in high temperature applications will be able to use our results and data to apply their own filtering.

That being said, our data can well be used to include a lower temperature criteria if high temperature operation is targeted.

Additionally, in replying to R#1 about model benchmarking, we assessed our model further at different temperature. In the revised manuscript, we have updated Table 2 to include the statistics for a lower annealing temperature. Relevant discussions have been appended in the results section.

3. On a practical note, despite maintaining a single phase, the combination of some HEAs with elements of very different melting points, vapor pressure, and oxidation resistance makes fabrication very difficult. This is pointed out by the authors themselves on the processing of the CoFeMnNiZn alloy. Furthermore, other aspects such as the cost or toxicity of a lot of metallic elements can also make the practical use of the alloys prohibitive.

We thank the reviewer for the insights on the practical use of alloys. Our present work is dedicated to the phase stability of multicomponent alloys, a fundamental aspect for HEA designs in view of the vast chemical space. While manufacturing challenges, cost and toxicity are certainly crucial factors to be taken into account for the widespread implementation of HEAs, we remain confident that our main point on the phase stability is not invalidated by these extra considerations. In addition, consideration like costs and toxicity are also very field dependent and the types of elements acceptable in electrocatalysis (e.g., noble metals) are not the same than in mechanical alloys. Our data can easily be used to in addition to phase stability add other considerations such as cost. As we have explained in previous replies we are trying to keep an application agnostic approach.

The reviewer suggests the authors include further screening criteria (e.g., mechanical properties and microstructure at target service temperatures) and verify them experimentally. In many cases, the criteria of single phase and equiatomic composition can be relaxed.

We refer to our reply above about adding additional screening.

Minor issues:

1. The introduction needs some rephrasing in that several phrases and arguments are inappropriate: Line 28 “disrupted”; Line 32 high configuration entropy does not seem to be the main reason for stable single phases (see <https://doi.org/10.1016/j.actamat.2016.08.081>); Line 33 only a very limited number of HEAs has shown promising properties; Line 73 limiting the interaction to binary terms needs to be justified. A discussion is needed in the main text or the supplementary material.

We have made the changes to the main text following the reviewer’s suggestions.

- The first sentence has been rephrased: “The field of metallurgy has been recently impacted by the emergence of high-entropy alloys”.
- To show the relevance of factors beyond configurational entropy, we have rephrased the sentence “The seemingly surprising stabilization of HEAs. . .” to “The seemingly surprising stabilization of multicomponent alloys against the formation of multiple phases and intermetallics (IMs) has been associated with the high configurational entropy among other important factors.”
- “HEAs have shown unusual distinguishing properties” has been changed to “HEAs

can exhibit unusual properties”.

- In the paragraph before the results section, “...for which the interactions are limited to binary terms” has been changed to “...for which the interactions are described by binary terms”. We restricted the interactions to the binaries as they are sufficient to reproduce the mixing enthalpy of quaternary and quinary solid solutions. This is explained at the beginning of the results section of our paper.

2. *A brief discussion of the possible effect of chemical short-range order (CSRO) is needed. A recent kinetic Monte-Carlo simulation (<https://doi.org/10.1016/j.commatsci.2021.110670>) shows that the time scale of forming atomic CSRO is on the order of microseconds to milliseconds, which means that CSRO can form during quenching.*

We thank the reviewer for pointing out the importance of SRO. The SRO effect is missing from our model as the mixing enthalpy is solely modeled with random solid solutions. We have indicated this limitation in the discussion section and cited the kinetic MC simulation accordingly.

3. *The usefulness of the statistical analysis given in figures 3 and 4 is limited because they give little information in terms of the design of a specific HEA. It is advised that the authors put them into the supplementary material.*

We respectfully disagree with the reviewer and consider both Figs. 3 and 4 important to show the important driving factors for high entropy alloy formation. In view of the comment from R#2 that the reader is “easily overwhelmed by the sheer number of stable single-phase high entropy alloys”, Figs. 3 and 4 and the text is there to help the understanding of why these combinations of elements appear among the hundreds of thousands of possibilities.

4. *The paragraph from Line 217-228 gives little information and implies strongly that the understanding of quinary HEAs relies on case-by-case scrutiny, which does not help the idea of high throughput screening.*

The paragraph in question is the following: “The milder effect of the mixing enthalpy can be surprising at first but is rationalized by its correlation with intermetallic formation. Elements that tend to mix strongly as a solid solution are also more

likely to form ordered intermetallic phases that in turn destabilize the solid solution. This was hinted previously by Senkov and Miracle. Taking all binary systems from the combination of the 40 elements, we explicitly show the linear correspondence between the formation enthalpy of intermetallics and the mixing enthalpy of solid solutions in Fig. 4. Following Fig. 1, we build a matrix indicating the competition between intermetallics and solid solutions for any given pair of the 40 elements. Interestingly, the Cantor alloy contains elements that tend to mix mildly together, not too favorably, not too unfavorably. While this could seem at first sight to be detrimental for HEA formation, this mild mixing in solid solution correlates with weak competition from intermetallics.”

We respectfully disagree that this paragraph gives little information. The correlation between intermetallic and solid solution formation is important as it mitigates the idea of favoring simply elements mixing favorably (as they will also form competing intermetallic easily). We also rationalize why among all possible elements the Cantor alloy is forming an equimolar quinary solid solution. We think these are important findings.

5. The authors need to describe how representative figures 6 and 7 are (maybe they belong to a small equiaxed grain region surrounded by columnar grains with large segregation) and provide the quantitative EDS results.

Additional information including SEM micrograph at lower magnification as well as EDX measurements are now provided in the supplementary information. This change is reflected in the main manuscript.

Response to Reviewer #3

The paper, “A map of single-phase high-entropy alloys”, describes a new model, developed using DFT databases, to predict whether high-entropy alloys (HEAs) across a wide range of compositions will be stable. The paper also describe the successful experimental synthesis of two alloys predicted by the model to be stable. Overall, the paper is well written and describes a valuable contribution to the study of high-entropy alloys (HEAs). The paper describe the model clearly and includes a useful discussion of various results from the model. I recommend the paper for publication, and provide the following com-

ments that may help the authors clarify some of the presentation of results and provide suggestions for some additional discussion:

We appreciate the positive recommendation from the reviewer. We respond to each of the comments in what follows.

Comments:

- In the introductory paragraph the authors include the sentence "The seemingly surprising stabilization of HEAs against 31 the formation of multiple phases and intermetallics has been credited to the high configurational entropy of multicomponent solid solutions." Ref. 3 cited here (D. Miracle and O. Senkov, A critical review of high entropy alloys and related concepts, Acta Mater. 122, 448 (2017).), and the authors previous work in Ref. 48 (G. Bokas, W. Chen, A. Hilhorst, P. Jacques, S. Gorsse, and G. Hautier, Unveiling the thermodynamic driving forces for high entropy alloys formation through big data ab initio analysis, Scr. Mater. 202, 114000 (2021).) makes clear that the role of entropy is important, but that it is hard to clearly distinguish, especially across HEAs in general, in a way this sentence does not seem to capture. While that complexity cannot be fully described in this introduction, it would be useful to at least hint at it.

We thank the reviewer for bringing up the complexity of the driving forces towards HEA formation. This actually coincides with one of the points raised by R#2. In the revised manuscript, we have changed the sentence to make it clear that the HEA stability depends also on factors beyond configurational entropy.

- On pg. 5 the authors state "We assume no configurational entropy for these ordered phases." It seems like the effect of this approximation should be discussed more carefully in HEAs. See for example Ref. 3, sec. 2.2.2.

In Refs. 3 Sec.2.2.2, the intermetallic considered are multicomponent intermetallic phase (e.g., quaternaries) where indeed there could be some stabilization by configurational entropy through disorder on sublattices. We do not consider these intermetallics and only use binary (and ternary in the revised version) intermetallics for which neglecting configurational entropy is appropriate.

We have clarified the text and added a comment in the manuscript about the implication of this approximation.

- Particularly due to the Methods section being located at the end of the paper, I found that on first reading I had some trouble understanding what exactly various quantities in the figures referred to, whether because the calculation method was unclear or whether I was unsure if the data was experimental vs model predictions. Some examples: - In Fig. 1, "...formation energy for binary solid solutions obtained from DFT..." could be changed to "...formation energy for binary solid solutions, as represented by SQS, obtained from DFT..."

We thank the reviewer for the suggested changes. The caption of Fig. 1 has been made modified accordingly.

- In Fig. 2a, "Structural preference of binary solid solutions..." could be changed to "Predicted structural preference of binary solid solutions...", and in Fig. 2b, "Ground-state structures of binary solid solutions..." could be changed to "Predicted ground-state structures of binary solid solutions..."

We have implemented the proposed changes to the caption of Fig. 2.

Fig. 4 is a bit confusing. Using the singular "Enthalpy of formation (intermetallics)" instead of "Lowest intermetallic formation energy", or something similar, seems unnecessarily confusing. It also seems confusing to compare "Enthalpy of mixing (solid solution)" to the inset orange line with a single point from the SQS calculations instead of perhaps the curve calculated from the regular solution model. I understand the equivalence after reading through the entire paper and methods section, but it was not immediately clear when first viewing the figure.

We have modified the caption of Fig. 4 following the reviewer's suggestions. We have also made it clear that the solid solutions are equimolar as indicated by the convex hull subplot.

- On page 13, line 259, "Among the few alloys that are stable at low temperatures are a series of Bi and Ag alloys of low melting point" → "Among the few alloys that our model predicts to be stable below X% Tm are a series of Bi and Ag alloys..."

The suggested change has been implemented in the revised manuscript.

- FIG. S2. “Enthalpy of formation of binary solid solutions” → “Enthalpy of formation of binary solid solutions, obtained from DFT calculations of SQS, for...” or “Predicted enthalpy of formation of binary solid solutions for...”
- FIG. S3 and S5: Similar to S2.

The captions of the aforementioned figures (now Figs. S4, S5, and S7) have been changed to highlight that the results are predicted by our model.

- *Could the authors clarify the criteria for "strongly favoring" or "strongly disfavoring" mixing used to draw the lines in Fig. 1?*

In Fig. 1, the strongly favoring mixing (enclosed by red boxes) refers to a cluster of binaries with a mixing enthalpy lower than -0.2 eV/atom, and the strongly disfavoring mixing (by blue boxes) refers to those with a mixing enthalpy higher than 0.2 eV/atom. We have modified the caption to include these criteria.

- *Could the authors discuss the effect, if any, of validating the model on the prediction of experimentally found stable compositions but not on known unstable compositions?*

Our revised manuscript now includes 61 alloys that are known not to form single phase for model verifications. In addition, we consider temperatures that are more representative to the annealing process in order to give a more realistic assessment of the predictive power of various models. By considering both the true positive rate (the capability of the model to predict true single-phase HEAs) and the false positive rate (how likely the predicted single-phase HEAs are falsely classified as single phase), we find that our present model is still the most accurate and balanced among all the models considered in our work.

The part of model validations has been largely rewritten in the revised manuscript.

- *Did the authors attempt to synthesize any other compositions that did not form stable solid solutions?*

Not yet but we are actively working using this data (as we expect other groups will do) and we will probably have more example of false positive results in the future. As a confidential note to the reviewer and editor, we have kept on working on

some of the Zn-based alloys suggested by the model and two other Zn-based HEAs have been processed. Preliminary results showed that both compositions formed single phase FCC alloys. For instance, Figure 1 shows x-ray diffraction spectra of CoCrFeNiZn for different heat treatments showing FCC reflections. Meanwhile, our model indeed predicts that CoCrFeNiZn can be stabilized in the FCC solid solution at $0.9T_m$ with an inverse energy above hull of -14.6 meV/atom. We do not include the results in the main text as we believe that these results fall outside the scope of the present study. However, this shows the potential of the proposed model as an important tool for future research.

Figure 1: XRD spectra of CoCrFeNiZn after processing and subsequent heat treatments showing FCC reflections.

- *The ER models have more false predictions of unstable compositions in the set of 75 stable compositions than the authors model. Do the ER models also have fewer false predictions of stable compositions?*

Indeed, Table 1 shows that the ER models (except ER1) have higher miss rate (100 - true positive rate) than our present model. At the same time, they have a higher false positive rate compared to our model at the typical annealing temperature. ER1 for instance has a false positive rate of 80%, rendering the model ultimately unreliable for classification.

Places where small edits are necessary:

pg. 4, line 80: *“is a very important driving factor the formation of HEAs but identify. . .”*
→ *“is a very important factor driving the formation of HEAs, but identify. . .”*

pg. 5, line 96: *“can be fitted for instance on DFT.”* → *“can be fitted, for instance on DFT.”*

pg. 12, line 232: *“In addition, to the mixing effect discussed previously where. . .”* →
“In addition to the mixing effect discussed previously, where. . .”

pg. 13, line 253: *“The analyses hitherto are based on the hypothesis that the synthesis temperature can be as high as near melting point and that HEA alloys will be quenched to their operation temperature (e.g., room temperature).”* → *“The analyses hitherto assume that the synthesis temperature can be as high as 90% T_m and that. . .”*

All the proposed edits have been taken into account during revision. We thank the reviewer once again for the suggestions that help improve our manuscript.

Reviewers' Comments:

Reviewer #1:

Remarks to the Author:

In the revised version of the manuscript and in the response to the referees the authors have addressed the points raised satisfactorily. In view of the additions and clarifications made in the revised version, I believe the manuscript is suitable for publication.

Reviewer #2:

Remarks to the Author:

The reviewer recognizes that the submitted response to the review is well-written. If the reviewer understands the response correctly, the gist of the response is the following:

1. The scope of the paper is to search for equiatomic compositions that will form single-phase alloys in a vast compositional space, and the impact can be enlarged when combined with further detailed computations driven by desired properties. However, the latter requires a very different searching strategy and therefore is beyond the scope of the current paper. In addition, once an alloy system has been identified due to a combination of reasons (desired phase, melting point, cost, sensitivity to cracking, etc.), the same searching methodology can be applied to varying the alloy system away from the equiatomic composition.
2. A major impact of this paper is to answer the fundamental question of the true and general effect of quinary solid solutions given the highly coupled nature of different terms in thermodynamics. The larger database built upon this methodology will advance our fundamental understanding of the synergistic effects of different thermodynamic parameters that determine the phase stability of the given alloys.
3. The methodology used in this paper can be extended to different temperatures for property-driven design (e.g., 0.5-0.8T_m for space and aeronautical applications).

The paper can be benefited if the gist of the response is incorporated into the main text in either the introduction section or the discussion section to elucidate the scope, impact, and limitations of the paper.

A final minor comment on the paper is that the new supplementary EDS data are confusing. The color bars were not labeled, and the reviewer assumes that they are atomic percentages. The EDS maps along with the electron images do not have scale bars. In addition, the EDS maps look as if there was some very noticeable segregation within the grains. The reviewer is not sure whether the electron image was taken under SE or BSE mode, but it seems necessary to carry out some more careful EDS scans and better data presentation.

The reviewer will recommend this manuscript for publication if the above issues were resolved.

Reviewer #3:

Remarks to the Author:

I thank the authors for their revisions. I believe the inclusion of 61 alloys that are known not to form solid solutions in the validation, and the overall revisions for clarity further strengthen the paper. I respectfully disagree with reviewer #2 as to whether additional property predictions are necessary for this paper to have impact, and agree with the authors that the phase stability predictions of the model across a very wide composition space is an achievement of significant interest.

I recommend the revised manuscript for publication.

Response to Reviewer #1

In the revised version of the manuscript and in the response to the referees the authors have addressed the points raised satisfactorily. In view of the additions and clarifications made in the revised version, I believe the manuscript is suitable for publication.

We thank the reviewer for the positive recommendation.

Response to Reviewer #2

The reviewer recognizes that the submitted response to the review is well-written. If the reviewer understands the response correctly, the gist of the response is the following: 1. The scope of the paper is to search for equiatomic compositions that will form single-phase alloys in a vast compositional space, and the impact can be enlarged when combined with further detailed computations driven by desired properties. However, the latter requires a very different searching strategy and therefore is beyond the scope of the current paper. In addition, once an alloy system has been identified due to a combination of reasons (desired phase, melting point, cost, sensitivity to cracking, etc.), the same searching methodology can be applied to varying the alloy system away from the equiatomic composition.

2. A major impact of this paper is to answer the fundamental question of the true and general effect of quinary solid solutions given the highly coupled nature of different terms in thermodynamics. The larger database built upon this methodology will advance our fundamental understanding of the synergistic effects of different thermodynamic parameters that determine the phase stability of the given alloys.

3. The methodology used in this paper can be extended to different temperatures for property-driven design (e.g., $0.5-0.8T_m$ for space and aeronautical applications).

The paper can be benefited if the gist of the response is incorporated into the main text in either the introduction section or the discussion section to elucidate the scope, impact, and limitations of the paper.

We fully agree with the reviewer on these three points which we summarize the scope of our work, the capability of our model, and its prospect in computational designs targeting specific properties. We have incorporated these points in the discussion section of the revised manuscript.

A final minor comment on the paper is that the new supplementary EDS data are confusing. The color bars were not labeled, and the reviewer assumes that they are atomic percentages. The EDS maps along with the electron images do not have scale bars. In addition, the EDS maps look as if there was some very noticeable segregation within the grains. The reviewer is not sure whether the electron image was taken under SE or BSE mode, but it seems necessary to carry out some more careful EDS scans and better data presentation.

We have improved Supplementary Figs. S8 and S9. The EDS color bars have been labelled in atomic percentage (at. %). The scale bars have been added to all the micrographs. We have further indicated whether the micrographs is taken using secondary-electron (SE) or backscattered-electron (BSE) imaging.

The reviewer will recommend this manuscript for publication if the above issues were resolved.

We thank the reviewer again for the valuable suggestions.

Response to Reviewer #3

I thank the authors for their revisions. I believe the inclusion of 61 alloys that are known not to form solid solutions in the validation, and the overall revisions for clarity further strengthen the paper. I respectfully disagree with reviewer #2 as to whether additional property predictions are necessary for this paper to have impact, and agree with the authors that the phase stability predictions of the model across a very wide composition space is an achievement of significant interest.

I recommend the revised manuscript for publication.

We appreciate the positive feedback from the reviewer.